# Convergent Validity of the Edinburgh Postnatal Depression Scale and the Patient Health Questionnaire (PHQ-9) in Pregnant and Postpartum Women: Their Construct Correlations with Functional Disability

**DOI:** 10.3390/healthcare11050699

**Published:** 2023-02-27

**Authors:** Manit Srisurapanont, Awirut Oon-arom, Chawisa Suradom, Suchaya Luewan, Suttipong Kawilapat

**Affiliations:** 1Department of Psychiatry, Faculty of Medicine, Chiang Mai University, Chiang Mai 50200, Thailand; 2Department of Obstetrics and Gynecology, Faculty of Medicine, Chiang Mai University, Chiang Mai 50200, Thailand; 3Research Administration Section, Faculty of Medicine, Chiang Mai University, Chiang Mai 50200, Thailand

**Keywords:** data accuracy, depressive disorder, prenatal diagnosis, postpartum depression, psychosocial functioning, psychometrics

## Abstract

This study aimed to evaluate the convergent validity of the Edinburgh Postnatal Depression Scale (EPDS) and the Patient Health Questionnaire (PHQ-9) in Thai pregnant and postpartum women, using the 12-item WHO Disability Assessment Schedule (WHODAS) as the reference standard. Participants completed the EPDS, PHQ-9, and WHODAS during the third trimester of pregnancy (over 28 weeks in gestational age) and six weeks postpartum. The sample included 186 and 136 participants for the antenatal and postpartum data analyses, respectively. The antenatal and postpartum data showed moderate correlations between both the EPDS and the PHQ-9 scores and the WHODAS scores (Spearman’s correlation coefficients = 0.53–0.66, *p* < 0.001). The EPDS and PHQ-9 were moderately accurate in distinguishing disability (WHODAS score ≥ 10) from non-disability (WHODAS score < 10) in pregnant and postpartum participants, but the area under the curve of the PHQ-9 receiver operating characteristic curves in postpartum participants was significantly larger than that of the EPDS, with a difference (95% CI; *p*-value) of 0.08 (0.16, 0.01; *p* = 0.044). In conclusion, the EPDS and PHQ-9 are valid for assessing PND-related disability in pregnant and postpartum women. The PHQ-9 may perform better than the EPDS in distinguishing disability from non-disability in postpartum women.

## 1. Introduction

Perinatal depression (PND) is clinically significant depression that occurs during pregnancy or the first postpartum year. It is a common complication that often goes unrecognized and has devasting effects on mothers and infants. Studies estimate that 10–15% of pregnant and postpartum women may experience PND [1,2], with higher prevalence in low- to middle-income countries compared to high-income countries [3]. Without treatment, PND can have significant adverse long-term effects on both mothers and infants.

A standardized, validated tool for PND screening should be used in obstetric and postpartum care. While other screening tools have 20 items or more and need more than 5 min to complete, the 10-item EPDS and the 9-item PHQ-9 can be completed in less than 5 min [4]. Studies show that using either the EPDS or PHQ-9 to screen for depression in postpartum women is effective [5]. These questionnaires are both helpful in assessing PND depression in different clinical settings [6]. Both are reliable and accurate for postpartum depression assessment, but they seem to identify different behavioral symptoms of antenatal depression [7]. The PHQ-9 covers many physical symptoms, while the EPDS is better at detecting anxiety during the perinatal period.

The Edinburgh Postnatal Depression Scale (EPDS) is the most commonly used tool for PND case finding. In postpartum women, the EPDS has a sensitivity range of 0.60 (specificity 0.97) to 0.96 (specificity 0.45) for major depression only and from 0.31 (specificity 0.99) to 0.91 (specificity 0.67) for major or minor depression [8]. A meta-analysis of individual participant data collected from pregnant and postpartum women found that the combined sensitivity and specificity were highest, at a cut-off point of 11 or higher. However, to identify higher symptom levels of PND, a cut-off point of 13 or higher should be used [9]. Although the EPDS was primarily developed for screening postpartum depression, it also has high accuracy in detecting antenatal depression, with an optimal cut-off score of 11/12 and above, both in terms of sensitivity and specificity [10]. Previous findings suggested that the EPDS is a valid self-report for detecting postpartum depression in Thai women [11].

The Patient Health Questionnaire-9 (PHQ-9) is another commonly used tool for PND case finding. Wang and colleagues (2021) compared the PHQ-9 to a criterion standard psychiatric interview and reported that the standard PHQ-9 cut-off point ≥10 had high accuracy in detecting both antenatal and postpartum depression with a pooled sensitivity, specificity, and AUC of 0.84, 0.81, and 0.89, respectively [12]. Furthermore, they found that the receiver operating characteristics (ROC) curves of the PHQ-9 and EPDS were nearly identical, with a median correlation between the two of 0.59 and moderate categorical agreement. A study of Thai women living with HIV also found that PHQ-9 depressive symptoms in pregnant and postpartum women were associated with low quality of life [13]. These findings suggest that the PHQ-9 may be used as an alternative to the EPDS for PND case findings.

The PHQ-9 appears to have two major strengths and some drawbacks in use as a case-finding tool for PND. First, the PHQ-9 was developed from the diagnostic criteria for major depressive episode (MDE), defined by the Diagnostic and Statistical Manual of Mental Disorders (DSM). Therefore, its detection of PND would be very similar to the MDE, peripartum onset, which is a widely accepted diagnosis in clinical practice. Second, while the EPDS has several cut-off points for detecting PND, the PHQ-9 seems to have a consistent cut-off point of 10 for detecting clinically significant depression, which is also applicable for PND. Therefore, the PHQ-9 is relatively easy to use across healthcare settings, including the obstetric and postpartum care settings. Although there are some critics that the PHQ-9 items mainly focus on physical symptoms [7], most women can distinguish somatic symptoms related to their pregnancy (e.g., nausea in the first trimester, sleep difficulties related to caring for a newborn) from those related to depression [14].

Distinguishing (functional) disability from non-disability refers to the ability to identify whether individuals are experiencing a significant impairment in their ability to perform daily activities and tasks, or if they are functioning normally. For a broader concept, disability includes the areas as follows: (1) understanding and communication, (2) self-care, (3) mobility, (4) interpersonal relationships, (5) work and household roles, and (6) community and civic roles or participation [15]. In clinical practice, this distinguishing has two-fold benefits. First, disability is a crucial characteristic for diagnosing mental disorders, including PND [16,17]. Furthermore, individuals who are experiencing a high level of disability should be given priority for intervention or support.

Convergent validity is a method to evaluate the accuracy of a case-finding tool (for diagnosis) by examining the correlation between their scores and the scores from other instruments that measure the same construct [18]. This is done through “hypothesis testing”, which involves determining whether the correlation between the scores is what would be expected. If the correlation is as expected, the case-finding tool is considered to have good convergent validity. Based on the hypotheses mentioned above, pregnant and postpartum women with more severe depression should also have more severe disability.

The EPDS and PHQ-9 scores, commonly used to measure the severity of PND, should be correlated with disability. So far, two studies have used the WHO Disability Assessment Schedule (WHODAS) as a reference standard for testing the convergent validity of EPDS and PHQ-9 against disability. The EPDS and PHQ-9 scores collected from Ghana postpartum women were weakly correlated with the WHODAS scores (r = 0.19 and r = 0.22, respectively) [19]. The antenatal PHQ-9 scores were weakly to moderately correlated with the WHODAS scores (r = 0.38 in Ghana and r = 0.41 in Côte d׳Ivoire) [20]. These findings suggest that the EPDS and PHQ-9 scores may not effectively measure disability in PND, raising questions about their use as diagnostic tools for PND, which usually needs a crucial characteristic of disability.

Recently, we conducted a study to investigate the use of biological markers in predicting antenatal and postpartum depression in Thai pregnant women [21]. In the current study, we used the EPDS, PHQ-9, and 12-item WHODAS scores from that previous study to perform a secondary data analysis to determine the convergent validity of EPDS and PHQ-9 in Thai pregnant and postpartum women by using the WHODAS as a reference standard. Additionally, we aimed to evaluate the overall performance of both questionnaires in distinguishing disability from non-disability in these populations.

## 2. Materials and Methods

### 2.1. Participants and Study Design

This study was approved by the Ethics Committee for Human Research, Faculty of Medicine, Chiang Mai University (Approval Number 367/2518) and adhered to the Declaration of Helsinki (1964) and its subsequent revisions. It was a single-center, prospective, observational study conducted at Maharaj Nakorn Chiang Mai Hospital, a public hospital at the tertiary level in Chiang Mai, Thailand. We approached adult participants during their first visits in the third trimester of pregnancy. Participants provided informed consent after the study details were fully explained. The study took place between December 2018 and November 2019. Because the materials and methods were fully described in our previous publication [21], we present only the highlights here.

The participants were a convenient sample of pregnant women aged 18–55 years who attended antenatal clinics for low- and high-risk pregnancies. They were screened for exclusion of psychotic disorders, current major depressive disorder, and dysthymia in the third trimester using the Mini International Neuropsychiatric Interview (M.I.N.I.) Thai version 5.0.0 [22,23]. We also excluded participants currently taking antipsychotics or antidepressants.

### 2.2. Assessing Depression, Anxiety, and Disability

Participants completed the EPDS, the PHQ-9, and the 12-item WHODAS questionnaire (version 2.0) in their Thai versions during the third trimester of pregnancy (more than 28 weeks in gestational age) and six weeks postpartum, [15,24,25,26,27]. The EPDS is a ten-symptom questionnaire that assesses depression over the week before the assessment. The PHQ-9 questionnaire uses nine symptoms to diagnose DSM-IV major depressive disorder. Each item on the EPDS or PHQ-9 is scored from 0 to 3 based on the severity of the symptom. The EPDS and PHQ-9 item scores are then summed to derive full scales of 0–30 and 0–27, respectively. The EPDS and PHQ-9 are reliable and valid measures for PND. However, they seem to assess different behavioral symptoms of depression [7]. While the PHQ-9 covers a broader range of physical symptoms, the EPDS tends to identify anxiety symptoms specifically during the perinatal period.

The WHODAS questionnaire assesses six dimensions of disability: (1) understanding and communication, (2) self-care, (3) mobility, (4) interpersonal relationships, (5) work and household roles, and (6) community and civic roles or participation. Each domain includes two questions, each using a five-level scale from 0, denoting “no difficulty”, to 4, denoting “extreme difficulty or cannot do”. The severity of each item is rated from 0 (none) to 4 (extreme or cannot do), and the scores of all items are summed to produce a total score from 0 to 48. Based on previous findings [28], a 12-item WHODAS score of 10 or more is considered clinically significant disability.

The 21-item Depression Anxiety Stress Scale (DASS-21), in its Thai version, was administered once during the third trimester of pregnancy [29]. The DASS-21 total score ranges from 0 to 63. Additionally, the DASS-21 includes three subscale scores that reflect the severity of depression, anxiety, and stress.

### 2.3. Statistical Analysis

The data collected from pregnant and postpartum participants were analyzed separately. We presented continuous and categorical data as mean (standard deviation, SD) and n (%). The internal consistencies of the EPDS, PHQ-9, and WHODAS were evaluated using Cronbach’s alphas, with a value of 0.8 and above considered good internal consistency [30]. The EPDS, PHQ-9, and WHODAS scores were considered ordinal data. Therefore, the Wilcoxon Signed-Rank test was applied to determine the score changes from the third trimester to the postpartum period, and the Spearman’s test was used for assessing the correlations among these scores.

We conducted convergent analysis using two methods. First, we reportedthe correlation between the severity of PND assessed using the EPDS or PHQ-9 and the disability measured using the WHODAS as Spearman’s correlation coefficients (r_s_) (95% confidence intervals, 95% CIs). The 95% CI of r_s_ was calculated based on the R distribution 95% confidence limits, which took into account the skewness and kurtosis of the sample data and provided a more accurate result. For the interpretation, the r_s_ values were categorized as follows: very strong (0.90–1.00), strong (0.70–0.89), moderate (0.40–0.69), weak (0.10–0.39), and negligible (0.00–0.10) [31]. Second, we plotted receiver operating characteristic (ROC) curves to determine the overall performance of EPDS and PHQ-9 scores in discriminating disability (WHODAS score ≥ 10) from non-disability (WHODAS score < 10). The area under the curve (AUC) of each ROC was calculated, and its accuracy was classified as poor (0.5–0.7), moderate (0.7–0.9), and high (>0.9) [32]. The AUCs EPDS and PHQ-9 ROC curves were compared using a paired design. A *p*-value less than 0.05 was considered statistically significant. All analyses were conducted using NCSS 21.0 [33].

## 3. Results

### 3.1. Antenatal Data Analysis

Out of 200 participants, 186 completed the EPDS, PHQ-9, and WHODAS in their third trimester of pregnancy and were included in the antenatal data analysis. The mean years of age and education were 29.46 (SD = 5.16) and 14.06 (SD = 3.37) years, respectively (see Table 1). Table 1 presents other characteristics of 186 pregnant participants included in the antenatal data analysis.

For the antenatal data (N = 186), the mean scores (SDs) of the EPDS, PHQ-9, and WHODAS were 6.35 (3.86), 4.38 (3.31), and 6.79 (5.88), respectively. Forty-eight participants (25.81%) had a clinically significant disability (WHODAS ≥ 10). The EPDS, PHQ-9, and WHODAS had good internal consistency, with Cronbach’s alphas of 0.80, 0.83, and 0.88, respectively. Figure 1 illustrates the correlation plots among the EPDS, PHQ-9, and WHODAS scores. The EPDS and the PHQ-9 scores were significantly and strongly correlated, with an r_s_ (95% CI; *p*-value) of 0.73 (0.63, 0.78; *p* < 0.001). The WHODAS score was significantly and moderately correlated with the EPDS score, with an r_s_ (95% CI; *p*-value) of 0.60 (0.50, 0.68; *p* < 0.001). The WHODAS score was significantly and moderately correlated with the PHQ-9 score, with an r_s_ (95% CI; *p*-value) of 0.56 (0.45, 0.65; *p* < 0.001). The ROC curves for the EPDS and PHQ-9 scores showed that both questionnaires had moderate accuracy in distinguishing disability from non-disability, with AUCs (95% CIs; *p* values) of 0.82 (0.73, 0.88; *p* < 0.001) and 0.79 (0.71, 0.86, *p* < 0.001), respectively (see Figure 2). The AUCs of both questionnaires were not significantly different, with a difference of 0.03 (95% CI -0.05, 0.10; *p* = 0.513).

### 3.2. Postpartum Data Analysis

Out of 200 participants, 136 completed the EPDS, PHQ-9, and WHODAS at the six-week postpartum visit and were included in the analysis of postpartum data. The mean years of age and education were 29.77 (SD = 5.34) and 14.02 (SD = 3.34) years, respectively (see Table 1). The mean scores (SDs) of the EPDS, PHQ-9, and WHODAS during the third trimester of pregnancy were 6.20 (3.61), 4.13 (2.82), and 6.83 (5.89), respectively. Maternal and newborn complications occurred in 51 (38.64%) and 15 (11.36%) of the participants. Table 1 presents other characteristics of the 136 postpartum participants included in the analysis of postpartum data.

For the postpartum data (N = 136), the mean scores (SDs) of the EPDS, PHQ-9, and WHODAS were 5.83 (3.99), 3.62 (3.50), and 5.27 (5.37), respectively. Twenty-five participants (18.38%) had a clinically significant disability (WHODAS ≥ 10). The EPDS, PHQ-9, and WHODAS had good internal consistency, with the Cronbach’s alphas of 0.82, 0.85, and 0.86, respectively. Figure 3 illustrates the correlation plots among the EPDS, PHQ-9, and WHODAS scores. These questionnaires were significantly and moderately correlated, with an r_s_ (95% CIs; *p*-values) as follows: (i) the EPDS and PHQ-9 scores = 0.66 (0.55, 0.74; *p* < 0.001), (ii) the EPDS and WHODAS scores = 0.53 (0.39, 0.64, *p* < 0.001), and (iii) the PHQ-9 and WHODAS scores = 0.57 (0.45, 0.68; *p* < 0.001). The ROC curves for the EPDS and PHQ-9 scores showed that both questionnaires had moderate accuracy in distinguishing disability from non-disability, with AUCs (95% CIs; *p* values) of 0.82 (0.70, 0.89; *p* < 0.001) and 0.90 (0.81, 0.94, *p* < 0.001), respectively (see Figure 4). The AUC of PHQ-9 was significantly larger than that of EPDS, with a difference of 0.08 (95% CI 0.16, 0.01; *p* = 0.044).

## 4. Discussion

The present findings add another dimension of evidence to the widely accepted questionnaires for screening PND. Their correlations with functional disability during perinatal periods suggest that the EPDS and PHQ-9 are valid for assessing PND in Thai pregnant and postpartum women. As disability is another dimension for diagnosing mental illnesses, these good correlations with disability indicate that a high EPDS or PHQ-9 score can be used to support the diagnosis of PND. Both questionnaires also have good internal consistency and are strongly correlated with each other. The scores of both questionnaires show moderate correlation with functional disability and moderate accuracy in distinguishing disability from non-disability. However, the PHQ-9 may be more effective than the EPDS in distinguishing disability from non-disability in postpartum women. Our findings should not be interpreted as the PHQ-9 being superior to the EPDS in screening for postpartum depression, due to limitations of the study. However, the PHQ-9 may still be considered as a viable alternative to the EPDS for this purpose in pregnant and postpartum women.

The antenatal data of this study showed more positive results on the correlations between the EPDS or the PHQ-9 scores and the WHODAS score compared to those reported in previous studies. Barthel et al. (2015) reported that the PHQ-9 score was weakly correlated with the WHODAS score in South African pregnant women (r = 0.38–0.41) [20], but our findings suggested that they were moderately correlated in Thai pregnant women (r_s_ = 0.56). Although it might be difficult to explain the differences between the previous and the present findings, it should be noted that the participants in the study of Bartherl et al. (2015) seemed to be more depressed, with mean PHQ-9 scores of 7.53 and 7.85, compared to our participants (mean PHQ-9 score of 4.38). The moderate correlation between the EPDS and the WHODAS scores found in this study (r_s_ = 0.60) has never been reported.

The postpartum data of this study showed that the EPDS and the PHQ-9 scores were moderately correlated with the WHODAS score in this population. In postpartum women, we found stronger correlations between the EPDS and WHODAS scores (r_s_ = 0.50) and between the PHQ-9 and WHODAS scores (r_s_ = 0.60) compared to only weak correlations in previous research (r = 0.19 and r = 0.22, respectively) [19]. While our study had a lower mean EPDS score (5.8) compared to the previous study (7.7), the mean PHQ-9 scores were similar in both studies (3.6). The difference in EPDS scores may explain the different correlations between the EPDS and WHODAS scores reported in previous and current studies. The discrepancy in findings may suggest that African and Asian postpartum women have different symptom profiles of postpartum depression and/or different perceptions of disability due to the fact that the PHQ-9 includes more somatic questions than EPDS.

To our knowledge, this study is the first to compare the overall performance of the EPDS and PHQ-9 in distinguishing disability from non-disability. The scores of both questionnaires show moderate accuracy in such differentiation. Our findings suggest that both questionnaires can be used to determine if a pregnant or postpartum woman is experiencing a level of disability. The larger AUC of PHQ-9 in postpartum women indicates that the PHQ-9 may be superior to the EPDS in distinguishing disability from non-disability in this population. Future studies are needed to identify the appropriate cut-off point of the PHQ-9 that reflects perinatal depression with clinically significant disability.

This study had some limitations. First, the small sample caused a low prevalence of participants with high symptom levels and/or clinically significant disability. The AUC difference between the EPDS and the PHQ-9 found in this study should be viewed only as preliminary findings and applied in clinical decisions with caution. The statistically nonsignificant associations or differences might be caused by type-II errors. Second, this is a single-site study in Thai women living in Chiang Mai and nearby regions. The effects of national circumstances, geography, and economic development should be considered. Therefore, these findings should be generalized to other populations with caution. Third, the clinically significant disability indicated by the WHODAS cut-off point of ≥10 was based on a population sample of Australian adults. Whether this cut-off point is appropriate for Thai adults, including pregnant and postpartum women, is unknown. Fourth, several factors that might affect the PND and the disability in this study population were not collected and taken into account, e.g., socioeconomic status, the severity of morning sickness, and the burden of newborn care. Last, as a study conducted during the COVID-19 pandemic, COVID-19 worries might affect the participants’ mood assessed using the EPDS and the PHQ-9. However, COVID-19 infection affecting mood symptoms would be low because the prevalence of COVID-19 infection was relatively low in Thai pregnant women [34]. Another issue of concern is the changes in living, socioeconomic, and environmental conditions after the COVID-19 epidemic. These problems may affect maternal mood differently and have to be considered in generalizing the present findings.

Our findings further support the use of EPDS and PHQ-9 as case-finding tools for PND, for both antenatal and postpartum depression. Previous studies suggested that both tools have high sensitivity and specificity. By using the convergent validity analysis, the present findings support that EPDS and PHQ-9 scores are correlated with disability, a dimension needed for making a diagnosis of mental disorders. In addition, pregnant and postpartum women with high EPDS or PHQ-9 scores should be a priority group for treatment initiation because they are likely to have PND-related disability. Both questionnaires display strong internal consistency and a high degree of correlation with each other, which suggests that they are relatively comparable in measuring the severity of PND. Due to several limitations, the present findings cannot be taken as evidence that the PHQ-9 is better than the EPDS in screening for PND. Nonetheless, the results suggest that the PHQ-9 may not be inferior to the EPDS and may provide support for using it as an alternative tool to the EPDS in evaluating PND in pregnant and postpartum women. Our findings on the convergent validity further support the use of EPDS or PHQ-9 in clinical settings, because these questionnaires not only can identify women with PND but also can identify those with disability, who desperately need treatment.

## 5. Conclusions

The EPDS and PHQ-9 are valid for assessing PND-related disability in Thai pregnant and postpartum women. The scores of these questionnaires reflect not only the severity of PND but also the disability levels. Although the PHQ-9 may perform better than the EPDS in distinguishing disability from non-disability in postpartum women, these findings should be interpreted with caution due to several limitations of this study. The PHQ-9 may be an alternative to EPDS for assessing depression in pregnant and postpartum women. Our findings further support the use of EPDS or PHQ-9 in clinical settings. Future studies are needed to compare other psychometric properties of the PHQ-9 with those of the EPDS.

## Figures and Tables

**Figure 1 healthcare-11-00699-f001:**
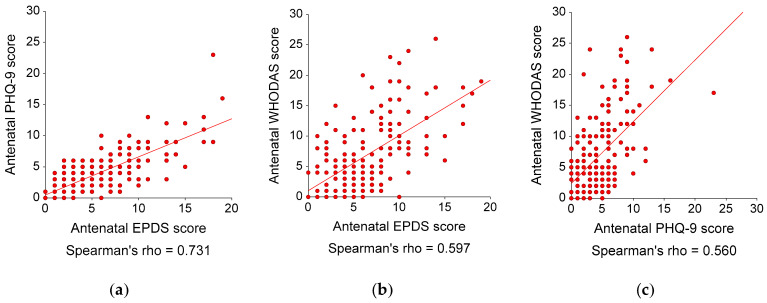
Correlations among EPDS, PHQ-9, and WHODAS scores and regression lines in 186 pregnant participants (antenatal data only): (**a**) Correlation between antenatal EPDS and PHQ-9 scores: Spearman’s rho (95% CI; *p*-value) = 0.73 (0.63, 0.78; *p* < 0.001; (**b**) Correlation between antenatal EPDS and WHODAS scores: Spearman’s rho (95% CI; *p*-value) = 0.60 (0.50, 0.68; *p* < 0.001); and (**c**) Correlation between antenatal PHQ-9 and WHODAS scores: Spearman’s rho (95% CI; *p*-value) = 0.56 (0.45, 0.65; *p* < 0.001). EPDS, Edinburgh Postnatal Depression Scale; PHQ9, 9-item Patient Health Questionnaire; WHODAS, 12-item World Health Organization Disability Assessment Schedule.

**Figure 2 healthcare-11-00699-f002:**
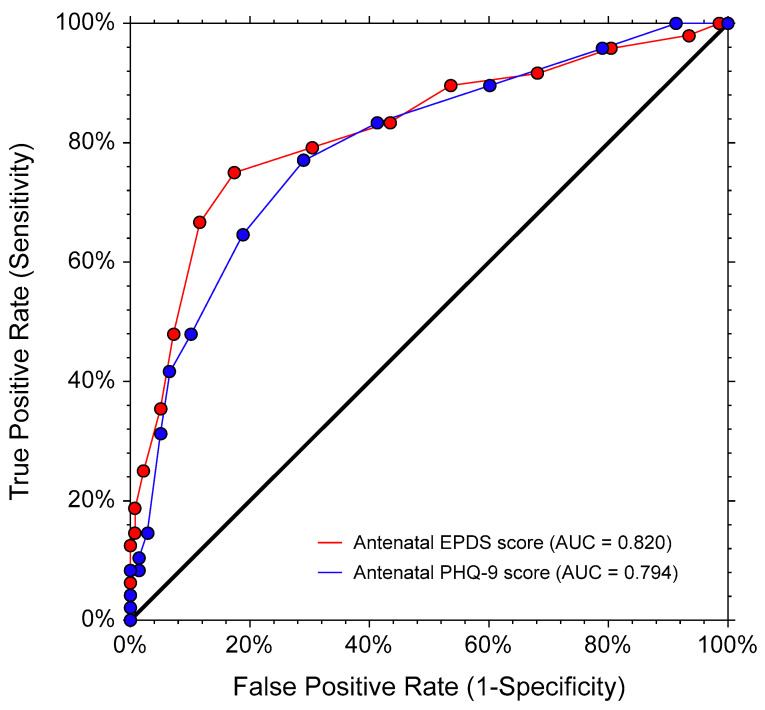
The receiver operating characteristic (ROC) curves of EPDS and PHQ-9 scores to differentiate disability (WHODAS score ≥ 10) from non-disability in 186 pregnant women (antenatal data only). AUC (95% CI; *p*-value): EPDS score = 0.82 (0.73, 0.88; *p* < 0.001), PHQ-9 score = 0.79 (0.71, 0.86; *p* < 0.01). AUC difference (95% CI; *p*-value): 0.03 (−0.05, 0.10; *p* = 0.513). AUC, area under the (receiver operating) curve (empirical estimate); EPDS, Edinburgh Postnatal Depression Scale; PHQ9, 9-item Patient Health Questionnaire; WHODAS, 12-item World Health Organization Disability Assessment Schedule.

**Figure 3 healthcare-11-00699-f003:**
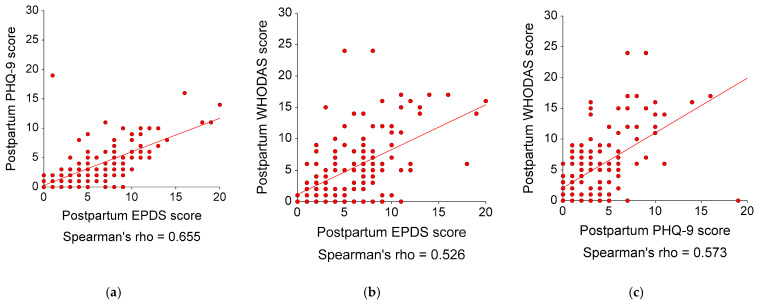
Correlations among EPDS, PHQ-9, and WHODAS scores and regression lines in 136 postpartum participants (postpartum data only): (**a**) Correlation between postpartum EPDS and PHQ-9 scores: Spearman’s rho (95% CI; *p*-value) = 0.66 (0.55, 0.74; *p* < 0.001); (**b**) Correlation between postpartum EPDS and WHODAS scores: Spearman’s rho (95% CI; *p*-value) = 0.53 (0.39, 0.64, *p* < 0.001); and (**c**) Correlation between postpartum PHQ-9 and WHODAS scores: Spearman’s rho (95% CI; *p*-value) = 0.57 (0.45, 0.68; *p* < 0.001). EPDS, Edinburgh Postnatal Depression Scale; PHQ9, 9-item Patient Health Questionnaire; WHODAS, 12-item World Health Organization Disability Assessment Schedule.

**Figure 4 healthcare-11-00699-f004:**
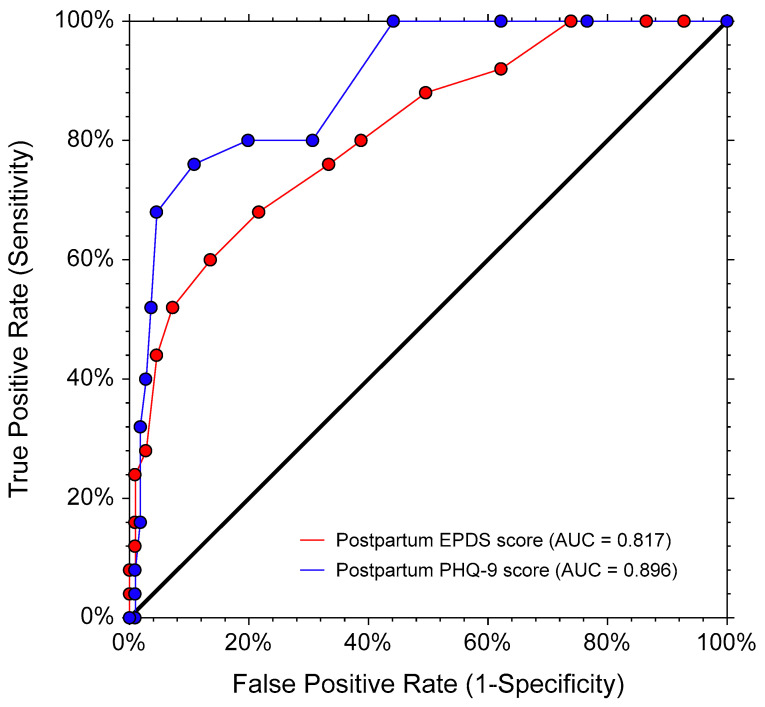
The receiver operating characteristic (ROC) curves of EPDS and PHQ-9 scores to differentiate disability (WHODAS score ≥ 10) from non-disability in 136 postpartum participants (postpartum data only). AUC (95% CI; *p*-value): EPDS = 0.82 (0.70, 0.89; *p* < 0.001), PHQ-9 = 0.90 (0.81, 0.94; *p* < 0.001). AUC difference (95% CI; *p*-value): 0.08 (0.16, 0.01; *p* = 0.044). AUC, area under the (receiver operating) curve (empirical estimate); EPDS, Edinburgh Postnatal Depression Scale; PHQ9, 9-item Patient Health Questionnaire; WHODAS, 12-item World Health Organization Disability Assessment Schedule.

**Table 1 healthcare-11-00699-t001:** Characteristics and psychopathology of 186 pregnant and 136 postpartum participants.

Characteristics	Pregnant Participants (N = 186)	Postpartum Participants (N = 136)
*At the third-trimester visit during pregnancy*	*Mean (SD)*	*Mean (SD)*
Age (years)	29.46 (5.16)	29.77 (5.34)
Education (years)	14.06 (3.37)	14.02 (3.34)
Body mass index (kg/m^2^)	23.03 (4.46)	23.23 (4.73)
DAS-depression score	4.76 (5.20)	4.23 (4.41)
DAS-anxiety score	5.29 (5.08)	4.94 (4.67)
DAS-stress score	6.89 (6.97)	6.76 (6.48)
Gestational number	1.75 (0.88)	1.75 (0.80)
Antenatal EPDS score	6.35 (3.86)	6.20 (3.61)
Antenatal PHQ-9 score	4.38 (3.31)	4.13 (2.82)
Antenatal WHODAS score	6.79 (5.88)	6.83 (5.89)
*At the third-trimester visit during pregnancy*	*n (%)*	*n (%)*
History of psychiatric disorders	6 (3.23)	5 (3.79)
Family history of psychiatric disorders	12 (6.45)	10 (7.58)
History of abortion	40 (21.51)	28 (21.21)
*Labor characteristics*		*n (%)*
Assisted delivery/Cesarian section		33 (25.00)
Maternal complications		51 (38.64)
Newborn complications		15 (11.36)
*6 weeks postpartum*		*Mean (SD)*
Postpartum EPDS score		5.83 (3.99)
Postpartum PHQ-9 score		3.62 (3.50)
Postpartum WHODAS score		5.27 (5.37)

DASS, 21-item Depression Anxiety Stress Scale; EPDS, Edinburgh Postnatal Depression Scale; PHQ9, 9-item Patient Health Questionnaire; WHODAS, 12-item World Health Organization Disability Assessment Schedule.

## Data Availability

The data is available upon request.

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
