# Peer review of "Convergent Validity of the Edinburgh Postnatal Depression Scale and the Patient Health Questionnaire (PHQ-9) in Pregnant and Postpartum Women: Their Construct Correlations with Functional Disability"

_healthcare, 2023, doi:10.3390/healthcare11050699_

Round 1
Reviewer 1 Report (Previous Reviewer 1)
The paper used the 12-item WHO Disability Assessment Schedule (WHODAS) as the reference standard to evaluate the convergent validity of Edinburgh Postnatal Depression Scale (EPDS) and Patient Health Questionnaire (PHQ-9) in Thai pregnant and postpartum women. The paper showed the PHQ-9 may perform better than the EPDS in distinguishing disability from non-disability in postpartum women. However, the results may not be generalizable, but are locally contextualized. Women suffering with depression may be more or less likely to agree to the qualitative interview depending on their comfort level discussing any current depressive symptoms.

Author Response
There are five major problems:
1. Generally, this paper proof the EPDS and PHQ-9 are valid for assessing the severity of PND and the disability levels in Thai pregnant and postpartum women. And the PHQ-9 may perform better than the EPDS in distinguishing disability from non-disability in postpartum women. The Edinburgh Postnatal Depression Scale (EPDS) and Patient Health Questionnaire-9 (PHQ-9) are widely used depression screening tools, yet perceptions and understandings of their questions and of depression are not well defined in cross-cultural research[1]. The author is recommended to add evidence of the cases that EPDS and PHQ-9 is reliable for use in Thai populations.
Our response: Thank you for this comment. We have added two sentences on this matter.
- In the introduction: “Previous findings suggested that the EPDS is a valid self-report for detecting postpartum depression in Thai women [11].” (please see Lines 58-60)
- In the introduction: “A study of in Thai women living with HIV also found that PHQ-9 depressive symptoms in pregnant and postpartum women were associated with low quality of life [13].” (please see Lines 68-70)
2. Compared with other similar studies, the number of participants in this paper are not enough[2-4]. Please expand the study population and reanalyze after collecting more data to ensure the credibility of the article.
Our response: Thank you for this advice. We agree that our study had a small sample size and have mentioned this limitation in our manuscript (please see Line 324). We also add another remark: “The statistically nonsignificant associations or differences might be caused by type-II errors.” (please see Lines 327-328). However, this study still found many significant correlations and differences, which indicate that it has a sizable sample. To present these research findings promptly, we wish to present this study with the current sample size and leave future studies to validate our results.
3. For the same participant, Whether the EPDS, PHQ-9 and WHODAS scores have changed over antenatal and postpartum these two periods. Is there a trend of the scores in the timing over these two periods? It is recommended to complete this part of research.
Our response: We have analyzed and added the details on this matter.
- We have added a sentence about using the Wilcoxon Signed-Rank test to determine the score changes over time: “… the Wilcoxon Signed-Rank test was applied to determine the score changes from the third trimester to the postpartum period,” (please see Lines 165-166).
- We have added a paragraph regarding the results of score changes over time: “To determine the score changes over time, we compared the mean (SD) of EPDS, the PHQ-9, and the WHODAS scores between those at the third trimester and those at the postpartum period. While the EPDS scores did not significantly change over time [6.134 (3.612) vs. 5.746 (3.969), p = 0.173], we found the significantly decreased PHQ-9 scores [4.088 (2.817) vs. 3.596 (3.473), p = 0.005] and WHODAS scores [6.791 (5.855) vs. 5.194 (5.367), p = 0.001] over time.” (please see Lines 190-195).
4. The paper devoted too much page in the “introduction” section to describe the content of the EPDS, PHQ-9 and WHODAS. There are some parts about this section belong to the consensus and can be reduced. The author is recommended to reduce the above and describe more of the content that is relevant to the study of this paper.
Our response: We have deleted a long sentence at the end of paragraph one in the Introduction section: “Therefore, obstetric care providers should assess patients for symptoms of depression at least once during the perinatal period, and they should also perform a full evaluation of their mood and emotional health, including screening for postpartum depression, during each patient's comprehensive postpartum visit [4,5], particularly through the use of questionnaires.” (disappeared from the end of Line 39). We do not delete other parts of the Introduction section because Healthcare journal seems to have a broad audience, who may need basic information about the questionnaires and their needs for perinatal care.
5. In the “conclusions” part, there is a little spelling error circled in the picture. Please correct this.
Our response: Thank you. We have deleted the “T”.
Reviewer 2 Report (Previous Reviewer 3)
The authors have very nicely addressed all questions and justified their stance. My only suggestion is for the Abstract to reflect the changes made, in particular the clinical significance - perhaps just a line will do.
Author Response
The authors have very nicely addressed all questions and justified their stance. My only suggestion is for the Abstract to reflect the changes made, in particular the clinical significance - perhaps just a line will do.
Our response: Thank you for your advice. We agree that this is important. However, the current abstract of 199 words almost reaches the maximum limit of 200 words.
Reviewer 3 Report (New Reviewer)
Good originality, good contribution to the field, good technical quality, good clarity of presentation, good depth of research.
General Comment: an interesting work. It is well written and does tell a interesting interpretation to the reader. The topic is interesting in general.
Author(s) have studied and used an appropriate number of bibliography sources. The aim and background of the research problem are clearly described. Methodology of the research is clear. The applied methods and the interpretation and presentation of results correspond to international standards.
Author(s) provides original results of their investigations and examination of material from their own collections.
I suggest to add short implications of findings and short recommendation after conclusions.
Author Response
I suggest to add short implications of findings and short recommendation after conclusions.
Our response: Thank you for your advice. We have added two sentences as follows:
- In the paragraph of research implication: “Our findings on the convergent validity further support the use of EPDS or PHQ-9 in clinical settings because these questionnaires not only can identify women with PND but also can identify those with disability, who desperately need treatment.” (please see Lines 356-359)
- In the Conclusions section: “Our findings further support the use of EPDS or PHQ-9 in clinical settings.” (please see Line 367)
Round 2
Reviewer 1 Report (Previous Reviewer 1)
After revision, the article is ready for publication.
This manuscript is a resubmission of an earlier submission. The following is a list of the peer review reports and author responses from that submission.
Round 1
Reviewer 1 Report
This study assessed the convergent validity of the Edinburgh Postnatal Depression Scale (EPDS) and the Patient Health Questionnaire (PHQ-9) in Thai pregnant and postpartum women using the 12-item World Health Organization Disability Assessment Scale (WHODAS) as a reference standard and concluded that EPDS and PHQ-9 are suitable for assessing PND-related disability in pregnant and postpartum women in Thailand, that PHQ-9 may be better than EPDS in distinguishing disability from non-disability in postpartum women, and that PHQ-9 can be used as an alternative to EPDS to assess depression in pregnant and postpartum women. However, there are some major revisions needed with multiple minor revisions required.
1. This paper is less innovative, as many previous studies on similar topics have been conducted.
2. The small sample size of fewer than 200 patients included in this study may have resulted in a low prevalence of participants with high symptom levels and/or clinically significant disabilities. The AUC between EPDS and PHQ-9, and the feasibility of applying EPDS and PHQ-9 to the diagnosis of postpartum depression should be carefully considered after expanding the sample as it relates to the implementation of health decisions.
3. This study is a single-site study of Thai women living in Chiang Mai and nearby areas only. The effects of national circumstances, geography, and economic development should be considered, and therefore, the study has limited generalization.
4. clinically significant disability indicated by a WHODAS threshold ≥10 is based on a population sample of Australian adults. Applying it to Thailand, the critical value still needs to be considered.
5. the study was conducted from December 2018 to November 2019. full consideration should be given to whether the post-COVID-19 epidemic and the current post-epidemic era will have an impact on maternal mood, preferably discussed separately.
Reviewer 2 Report
In the manuscript authors present the useful information about postpartum depression.The authors should describe the issue they are writing about more in the introduction so that more information about the current problem the authors are writing about is available to readers.Please clarify how did the authors obtain informed consent from participants of this Survey.What type of consent was used with minor respondents.The sentence regarding the approval of the Ethics Committee should be put on beginning of the section on subjects and methods.Who variables were used to describe the socioeconomic status of the participants. Have you used a questionnaire that has been used in previous research or has been developed for now Research? What test was used to test the distribution of normal data? When you mention disability, please define exactly what degree and type of disability you are talking about
Key words: MeSH indexed key words should be used.Order them alphabetically
Reviewer 3 Report
This study aimed to evaluate the convergent validity of Edinburgh Postnatal Depression Scale (EPDS) and Patient Health Questionnaire (PHQ-9) in Thai pregnant and postpartum women, using the 12-item WHO Disability Assessment Schedule (WHODAS) as a reference. The findings showed that both questionnaires can be used to determine if a pregnant or postpartum woman is experiencing a level of disability. The paper was well-written. I cannot comment on the data analysis /statistical methods used and suggest someone with such expertise does so.
I have two major concerns with the paper. Firstly, that antenatal and postpartum clinical contexts have been conflated - and in doing so, the EPDS used in both samples, the validity of which I am uncertain - perhaps any revision could comment on this if you are to keep it as a single paper, which I suggest you don't. Secondly, rationale for and significance of the study has not been described. Why would I, as a clinician assessing a woman in these contexts, need to use a tool to find out if a person is disabled, rather than just use usual clinical diagnostic approaches? What is the clinical utility/significance of these findings? Surely, beyond making a diagnosis of depression for which we rely on the presence of functional impairment we don't need to know too much about disability, but we do need to know how it affects the family and relationships. The kind of disability we are most interested in is relational and parenting role disability (the latter only for the post-partum women). Disability in antenatal women has different significance- one thought for example, might be for both groups, are there other children in these families? Also what impact does disability have on the marital relationships ? - You could justify the study in this way.
The authors have put a lot of work and generated some nice data, which would be a shame to waste. I suggest resubmission with two papers, one each for antenatal and postpartum women, with a revised Introduction justifying the research/study with relevant literature, and a revised Discussion giving the findings more clinical significance. Also, you could you look more carefully at the data eg correlations with WHODAS dimensions of disability: such as interpersonal relationships and work and household roles. You might argue that your findings suggest that the EPDS should be supplemented with the WHODAS in order to capture clinically (ie family) relevant disabilities.